# Transition-Age Young Adults with Cerebral Palsy: Level of Participation and the Influencing Factors

**DOI:** 10.3390/medicina55110737

**Published:** 2019-11-14

**Authors:** Zane Rožkalne, Maksims Mukāns, Anita Vētra

**Affiliations:** 1Rehabilitation Clinic, Children’s Clinical University Hospital, LV-1004 Riga, Latvia; anita.vetra@rsu.lv; 2Department of Doctoral Studies, Rīga Stradiņš University, LV-1007 Riga, Latvia; 3The Statistics Unit of Rīga Stradiņš University, LV-1046 Riga, Latvia; maksims.mukans@rsu.lv; 4Department of Rehabilitation, Rīga Stradiņš University, LV-1067 Riga, Latvia

**Keywords:** cerebral palsy, developmental transition, participation, disability, young adults

## Abstract

*Background and Objectives:* The aim of this study was to identify the level of participation in the context of the developmental transition from adolescence to adult life for young adults with cerebral palsy (CP) and the factors that had an influence on participation. *Materials and Methods:* Eighty-one young adults (16–21 years old) with CP and with normal or slightly decreased cognitive function participated in this study. Assessments were made using the Rotterdam Transition Profile (RTP) and the WHO Disability Assessment Schedule 2.0 (WHODAS 2.0). In the binary regression model, levels of participation (RTP scores) were set as dependents and the level of disability (WHODAS 2.0 scores), age, and level of gross motor function were set as independent variables. *Results:* In the age group <18 years, in three out of seven RTP domains, less than 10% of participants were in phase 2 (experimenting and orientating toward the future), i.e., finance—7%, housing—7%, sexuality—4%. In the age group ≥18 years, 21% (education and employment), 56% (intimate relationships), and 59% (sexuality) of the participants were in phase 0 (no experience). Higher scores in WHODAS 2.0 domains showed positive associations with RTP domains, i.e., cognition with social activities, mobility with transportation, self-care with sexuality and transportation, and life activities with transportation. Age was positively associated with education and employment, finance, housing, and sexuality. Low motor function according to the Gross Motor Function Classification System (GMFCS) had negative associations with autonomy in social activities, sexuality, and transportation. *Conclusions:* Young adults with cerebral palsy showed low levels of autonomy in all domains of participation. When addressing a person’s improvement in terms of their participation, the promotion of abilities in cognition, mobility, self-care, and life activities should be attempted. Age and gross motor function influenced autonomy in participation, but not in all domains.

## 1. Introduction

Cerebral palsy (CP) is the most common physical disability in children [1] and ranges in prevalence from 1.5 to 2.5 per 1000 live births [2]. It is a lifelong condition and the life expectancy in non-severe cases can be similar to the general population [3,4]. Therefore, much emphasis has now been put on the developmental and healthcare transition process from adolescence to adult life and participation [5,6,7,8,9]. It has been previously reported that young adults with CP participate less in activities, such as housing, paid work, and intimate and sexual relationships than their able-bodied peers [9,10,11], and in terms of finances and activities, many are dependent on parental support [12]. Participation has been defined as involvement in a life situation [13], and for patients with CP, enhancement of participation may lead to a more effective transition to adult life [14,15,16], whereas unemployment, decreased autonomy, and insufficient quality of life may be the consequences of an unsuccessful transition process [17,18]. Specific transition rehabilitation programs have been developed in European countries, such as the Netherlands [19] and the United Kingdom [20]. To our knowledge, there are no studies that present data about young adults with CP living in post-Soviet countries investigating their level of participation and/or transition process. Historically, healthcare-providing services have not been equally developed between different parts of Europe [21,22], and the identification of a specific situation in a concrete region is essential for the development of effective and purposeful rehabilitation programs. The aim of this study was to identify the level of participation in the context of the developmental transition from adolescence to adult life for young adults with CP living in Latvia and the influence of the level of disability, age, and gross motor function on their participation.

## 2. Materials and Methods

### 2.1. Study Design

The design of the study was non-experimental and cross-sectional in nature.

### 2.2. Setting and Population

Most of the participants where sought through the main child medical database in Latvia through the Children’s Clinical University Hospital paper records and electronic database. Open invitations were sent also to smaller youth and adult rehabilitation centers. The criteria for participants’ identification were: 16 to 21 years of age and diagnosed with CP (G80-G83, ICD-10). Those with cognitive impairment (score of Mini-Mental State Examination (MMSE) [23] of less than 24 points) and/or not capable of understanding the aim of the research and the process of the assessment, not really having a CP diagnosis, living in a social institution, emigrated from Latvia, not possible to gain contact with, not willing to participate, or dead were excluded. Before the assessment, participants or their legal representative (in cases of participants under the age of 18 years) received the informed consent and answers to questions, if any, concerning the study were put forward. After signing the informed consent, structured interviews using the assessment questionnaires were conducted. Assessments were carried out at a convenient place and time for the participants and were managed from October 2017 until June 2018. The investigations were carried out following the rules of the Declaration of Helsinki of 1975, which were revised in 2013. The study protocol was approved by the Ethics Committee of Rīga Stradiņš University (5/18.08.2016.) on 18 August 2016 and the Children’s Clinical University Hospital (SP-68/2016.) on 12 June 2016. This study forms part of the first author’s doctoral thesis.

### 2.3. Outcome Measures

The level of participation was measured with the Rotterdam Transition Profile (RTP) [11], version 1.0, March 2010. The tool is aimed at describing the transition process from childhood to adulthood for young adults with CP. In this study, we put more emphasis on participation in the context of the developmental transition and not on the healthcare transition; therefore, we used and analyzed only the first seven questions from the RTP as they measure the autonomy level in terms of participation; the remaining three are associated with healthcare. The level of autonomy is scored from 0 to 3 points, where 0 means “no experience” and 3 means “autonomy.”

The level of disability, as one of the influencing factors on participation, was measured with the World Health Organization Disability Assessment Schedule 2.0 (WHODAS 2.0) [24]. The tool consists of six domains: cognition, mobility, self-care, getting along, life activities (household and work), and participation. The disability level in each domain is measured by the difficulty of managing certain tasks described by the domain questions. It is a five-point Likert-scale system, where 1 means “no difficulty” and 5 means “extreme difficulty or cannot do.” The domain of participation was not included in this study as it conceptually overlaps with the RTP.

The level of gross motor function was measured with the Gross Motor Function Classification System (GMFCS) [25]. GMFCS is a five-level classification system, where level 1 refers to no functional limitations and total independence, and level 5 refers to the most severe limitations and high dependence.

### 2.4. Statistical Analysis

Statistical data analysis was performed using SPSS software (IBM SPSS Statistics, v. 23.0, Chicago, IL, USA). The study sample was tested for the normality of the distribution of the measurements using the Shapiro–Wilk test. As the sample did not disclose a normal distribution, non-parametric statistics were performed and quantitative data was analyzed in terms of the median (Me) and interquartile range (IQR). The Mann–Whitney U test was used for the comparison of WHODAS 2.0 domain values in the GMFCS level groups. GMFCS was dichotomized into groups with levels I and II versus III and IV. The Spearman rho method was used for the correlation of the interval, ordinal, and dichotomous data. The correlation analysis was performed with RTP participation domains and WHODAS 2.0, GMFCS, and age to find the strongest correlation coefficient. The interpretations of the correlation coefficient were as follows: up to |±0.19| very weak, |±0.20|–|±0.39| weak, |±0.40|–|±0.59| moderate, |±0.60|–|±0.79| strong, |±0.80|–|±0.999| very strong [26]. Binary logistic regression analysis was carried out with each of the influencing factors on RTP if the correlation was ≥|±0.40| for WHODAS 2.0 domains and ≥|±0.20| for GMFCS and age. A higher correlation coefficient for WHODAS 2.0 was used because there were some ideological question similarities with RTP questions. RTP data were divided into two groups according to phases 0 to 2 (totally to partly dependent) and phase 3 (autonomy), and were included in the binary regression model as dependent variable. Age, GMFCS, and WHODAS 2.0 were independent variables in the regression models. WHODAS 2.0 domains were dichotomized into two groups: ≤2.0 mean points (autonomy) versus >2.0 points (some level of a need of an assistance). Each of the WHODAS 2.0 domains’ sums were divided by its question count. The computed mean value ranged from 1 to 5 points. An OR (odds ratio) greater than 1 indicated larger odds of autonomy, while an OR smaller than 1 indicated smaller odds of autonomy. The significance level was *p* < 0.05. Figures were created in Microsoft Office Excel (2010) and tables were created in Microsoft Office Word (2010) (Redmond, WA, USA).

## 3. Results

A total of 225 potential participants were identified and, after the exclusion process, 81 took part in the study. The characteristics, including the functional levels of participants, are presented in Table 1. The median age for participants was 18 years, 51% were men, and two-thirds of the participants were 18 or more years old. The majority (74%) were highly functional (level I and II) in gross motor function according to the GMFCS. None of the participants were detected for GMFCS level V. A close to maximal or maximal score of the MMSE accounted for 59% of the participants.

Figure 1 shows the percentage distribution of participants’ regarding the transitional phases of each of RTP domain. The results have been divided according to the age of participants <18 years and ≥18 years.

In the age group <18 years, phase 2 was achieved by 11% of the participants (domain: education and employment), 7% (finance), 7% (housing), 26% (leisure (social activities)), 15% (intimate relationships), 4% (sexuality), and 15% (transportation). In the age group ≥18 years, phase 3 was achieved by 9% of the participants (education and employment), 26% (finance), 11% (housing), 57% (leisure (social activities)), 17% (intimate relationships), 24% (sexuality), and 44% (transportation); furthermore, 21% (education and employment), 56% (intimate relationships), and 59% (sexuality) of the participants in this age group were still in phase 0.

Table 2 demonstrates participants’ median values on the WHODAS 2.0 scale and total median scores. The median values are shown with participants being divided into two groups—more functional (GMFCS levels I and II) and less functional (GMFCS levels III and IV)—and for all participants without a division.

To elucidate the interactions between the two tools, we conducted a correlation analysis between RTP participation domains and WHODAS 2.0 disability-measuring domains (Table 3). The most frequent and strongest correlations with RTP participation domains were found with the WHODAS 2.0 self-care domain, i.e., housing: −0.42, intimate relationships: −0.44, sexuality: −0.46, and transportation: −0.75. The weakest correlations were with the WHODAS 2.0 getting along domain. For all moderate or higher (*r_s_* ≥ |±0.40|) correlations, the significance was *p* < 0.001.

To define the influencing factor’s (level of disability) associations with the autonomy level in the RTP domains (level of participation in the context of the transition), a binary regression analysis was performed (Table 4). The RTP domain was set as the dependent variable and the WHODAS 2.0 domains were set as the independent variables (only those with correlations of moderate or higher strength *r_s_* ≥ |±0.40|).

Logistic binary regression revealed that the WHODAS 2.0 cognition domain had associations with autonomy level in the RTP leisure (social activities) domain, i.e., OR = 8.1, 95% CI = 2.6–24.8. The mobility domain was associated with the RTP domain transportation (OR = 6.9, 95% CI = 2.5–18.7). The self-care domain was associated with two RTP domains—sexuality (OR = 9.4, 95% CI = 1.2–77.0) and transportation (OR = 53.3, 95% CI = 6.7–424.0)—but it did not reach the significance level in the RTP intimate relationships (*p* = 0.06) domain. The WHODAS 2.0 life activities domain was associated with the autonomy level in the RTP transportation domain (OR = 8.4, 95% CI = 3.0–23.3). The larger the OR, the higher the autonomy in the RTP domains.

Table 5 presents the correlations between the RTP domains and participants’ age and GMFCS level.

Participants’ age had weak correlations with the RTP domains of education and employment (0.26), finance (0.37), housing (0.31), and sexuality (0.28). The GMFCS level correlated with the RTP domains of leisure (social activities) (−0.26), intimate relationships (−0.26), sexuality (−0.31) (weak correlations), and transportation (−0.56) (moderate correlation). For all weak or higher (*r_s_* ≥ |±0.20|) correlations, the significance was, at least, *p* < 0.05. Further binary regression analysis was performed to find the influencing factors’ (age and GMFCS level) associations with autonomy level in the RTP domains (Table 6). The RTP domain was set as a dependent variable, and the age and GMFCS level as independent variables (only those with correlations of weak or higher strength *r_s_* ≥ |±0.20|).

Participants’ age was associated with autonomy in the following RTP domains: education and employment (OR = 2.4, 95% CI = 1.0–5.5), finance (OR = 2.1, 95% CI = 1.2–6.5), and housing (OR = 2.7, 95% CI = 1.2–6.5), which had the largest OR value, and on sexuality (OR = 1.6, 95% CI = 1.1–2.5). The larger the OR, the higher the participants’ autonomy in a specific RTP domain.

The GMFCS level did not meet the significance criterion regarding having an association with intimate relationships (*p* = 0.07). The smallest OR value of GMFCS was associated with transportation (OR = 0.2, 95% CI = 0.1–0.5), then with sexuality (OR = 0.3, 95% CI = 0.1–0.9), and following that, with leisure (social activities) (OR = 0.6, 95% CI = 0.4–1.0). The smaller the OR, the lower the autonomy.

## 4. Discussion

For young adults with CP living in Latvia, this is the first study that identifies their level of participation in the context of the developmental transition from adolescence to adult life and the influence of the level of disability, age, and gross motor function on the participation. To the best of our knowledge, this is the first study that explores the transition theme for young adults with cerebral palsy in one of the post-Soviet countries.

This study presents (Figure 1) the levels of participation in the context of the developmental transition and indicates the differences between minors (16–17 years) and adults (18–21 years). It has been previously shown that age is a significant factor when measuring readiness to transition to adult life [11]. By 18 years of age, individuals obtain almost full legal rights and more life opportunities; therefore, it might seem obvious that differences are found between minors and adults in the context of the transition process and of participation. However, in our study, we also found low scores of participation domains at the age of ≥18 years. Only 9% of adult age participants were in phase 3 in the education and employment domain (having “paid job, volunteer work”) and 44% were still in phase 1 in the finance domain (dependent on adults for “pocket money, clothing allowance”). Other studies also report that young adults with CP demonstrate low rates of employment and management of their finances [7,9,27,28]. As discovered by Verhoef et al. [9], in the age range of 20–24 years, young adults with CP have lower employment rates and higher unemployment rates than the general population. In our study, 78% of adult-aged participants in the housing domain were still in phase 1 (“living with parents, not responsible for household activities”) and little more than half, i.e., 57%, were in phase 3 (“young adult goes out in the evening with peers”) in the leisure (social activities) domain. Van der Slot et al. [28] revealed that at least 60% of adults with CP experienced difficulties with recreation and housing, as well as mobility. Our study shows that in the transportation domain, more than one-third, i.e., 37%, were still in phase 1 (“parents or caregivers transport the young adult”—completely dependent on parents or caregivers). The most severe inexperience (phase 0) for adult-aged participants was in the domains of intimate relationships and sexuality with 56% (“no experience with dating”) and 59% (“no experience with French kissing”), respectively. Wiegerink and colleagues [29] found that in the age range of 20–24 years, 45% of young adults with CP feel emotionally inhibited to initiate sexual contact, and for 90% of participants, sexuality was not discussed during the rehabilitation courses. Other research studies also emphasize the lack of experience and autonomy regarding this topic [10,27,30].

In our study, we demonstrated the levels of difficulty in managing health-related domains according to WHODAS 2.0 (Table 2). Participants were divided in two groups: less and more functional. In most cases, more functional participants showed higher scores regarding managing tasks at a significant level, i.e., mobility, self-care, life activities, and total score of WHODAS 2.0. Previous studies have also revealed that lower levels of functionality correlate with lower scores of activities associated with participation, e.g., employment [9], leisure activities and transportation [11], and self-care [6]. In our research, we did not find significant differences between less- and more-functional young adults with CP and the WHODAS 2.0 domains of cognition (understanding and communicating) and getting along (interacting with other people). This might be explained by the fact that, in our study, we had participants with none or some/uncertain cognitive impairment. However, the underlying aspects should be studied in greater detail.

In the correlation analysis between the RTP participation domains and WHODAS 2.0 domains, we found moderate to strong correlations between the RTP transportation domain and WHODAS 2.0 mobility, self-care, and life activities domains. According to WHODAS 2.0 domain descriptions [24], it can be assumed that the level of autonomy regarding the organization of their own transportation is associated with abilities in mobility (standing, moving around inside the home, getting out of the home and walking a long distance), self-care skills (bathing, dressing, eating, staying alone), and life activities, such as household, work, and school activities. Moderate to strong correlations were also found between the WHODAS 2.0 self-care domain and the RTP housing, intimate relationships, sexuality, and transportation domains. Self-care promotion is a crucial aim in rehabilitation [16], and as our study shows, it has strong associations with many important life and participation aspects. A logistic binary regression revealed that the WHODAS 2.0 domain of self-care was not significant in terms of its influence on the RTP domain of intimate relationships. This might be explained by the fact that the intimate relationships domain does not necessarily involve physical contact, as the sexuality domain (kissing and intercourse) does. Other influencing factors were also significantly associated with RTP domains, i.e., cognition on leisure, mobility on transportation, self-care on sexuality and transportation, and life activities on transportation. Those with higher scores of certain WHODAS 2.0 domains (>2.0 point mean value) were more likely to develop higher autonomy in specific RTP domains.

According to our findings, age had significant, but weak, correlations with the RTP domains of education and employment, finance, housing, and sexuality. A logistic binary regression with these domains showed that the older the participant became, the more autonomy he/she achieved. Schmid et al. [31] also found that with age, the level of autonomy in participation increased, with autonomy in sexuality being the last one to be achieved.

The GMFCS level significantly correlated with the RTP domains of leisure (social activities), intimate relationships, sexuality, and transportation. In contrast, a logistic binary regression revealed that the GMFCS level did not reach significance regarding the association with the RTP domain of intimate relationships. In our opinion, the possibilities of social media in the development of relationships might have a role in this [32], and if a person is not developing a physical contact, the level of gross motor function, at some point, may not yet exert an influence. Logistic binary regression shows that low gross motor function is associated with lower autonomy in leisure (social activities), sexuality, and transportation. Jacobson et al. [12] also revealed that for young adults with CP, their functional level has an influence on social participation aspects, such as socialization with friends and experience in intimate relationships (in this study, it was not categorized as being with or without sexual intercourse). Furthermore, Schmid et al. [31] found that the autonomy in transportation is lower for those GMFCS levels III–V.

Implications for practice that arise from this study are the following: (1) When preparing an adolescent with CP for adult life, rehabilitation teams should pay attention to participation aspects, such as education and employment, financial independence, household managing and independent living, leisure activities, intimate and sexual education, and autonomy in transportation. (2) An increase in self-care abilities should be emphasized when preparing the adolescent with CP for autonomy, especially in aspects such as sexuality and transportation. (3) When promoting autonomy in leisure activities, attention should be paid to the improvement of cognitive function (understanding and communicating). (4) Even though mobility and gross motor function improvement is one of the main outcomes in the rehabilitation for children with CP, it still remains a participation-limiting problem in young adults; therefore, more emphasis should be put on the promotion of mobility. (5) Age is a certain indicator for the level of autonomy, but for young adults with CP in the context of participation, it cannot be taken as the only indicator, especially in aspects such as education and employment, finance, and sexuality. (6) The findings of this study serve as a rationale for the need for transition rehabilitation in Latvia and they may potentially be similar in other post-Soviet countries.

The limitations of this study are as follows: we did not have able-bodied peers as a reference group, and as this is a cross-sectional study, it does not give information on changes over time. In future research, an in depth qualitative study regarding the level of participation for young adults with CP living in Latvia is planned.

## 5. Conclusions

In all domains of participation in the context of the developmental transition, young adults with cerebral palsy show low levels of autonomy. When addressing a person’s improvement in terms of participation, improvement of the abilities regarding cognition, mobility, self-care, and life activities should be attempted. Age has a positive influence on the level of participation in the context of the developmental transition, but not significantly on leisure (social activities), intimate relationships, and transportation. Lower levels of gross motor function show a significant negative influence on participation in the domains of leisure (social activities), sexuality, and transportation.

## Figures and Tables

**Figure 1 medicina-55-00737-f001:**
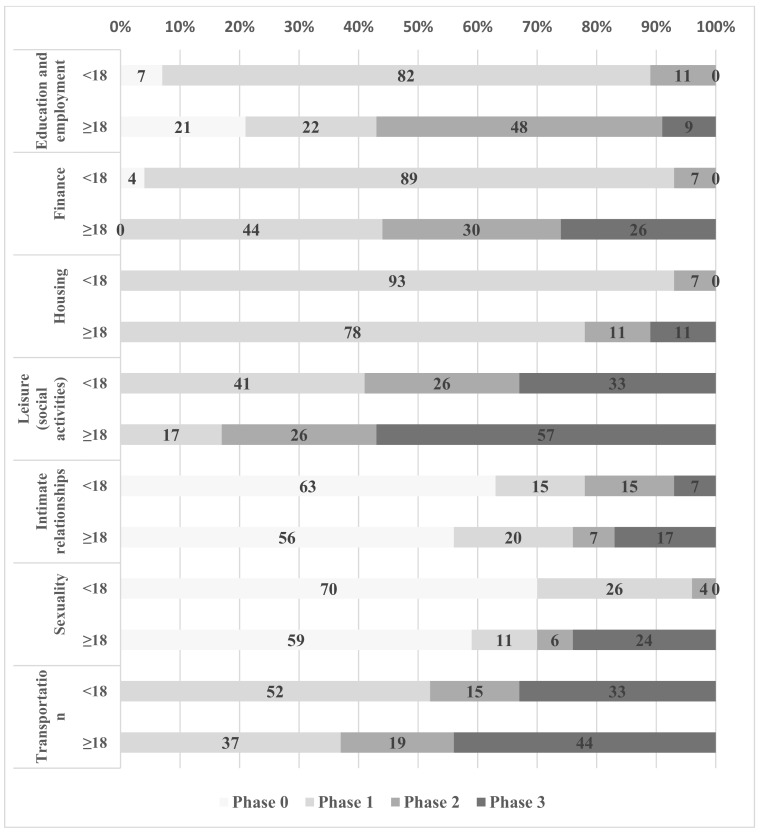
Rotterdam Transitional Profile (RTP) transitional phases of participants. Phases: 0—no experience, 1—dependent on adults, 2—experimenting and orientating with the future, 3—autonomy.

**Table 1 medicina-55-00737-t001:** Characteristics of the study participants (*n* = 81).

Age in years, Me (IQR)	18 (20–17)
Age distribution, *n* (%)	
<18 years	27 (33)
≥18 years	54 (67)
Gender, n (%)	
Men	41 (51)
Women	40 (49)
GMFCS, n (%)	
Level I	36 (44)
Level II	24 (30)
Level III	13 (16)
Level IV	8 (10)
MMSE, n (%)	
30 points	17 (21)
28–29 points	31 (38)
26–27 points	25 (31)
24–25 points	8 (10)

Me: median, IQR: interquartile range, CP: cerebral palsy, GMFCS: Gross Motor Function Classification System, MMSE: Mini-Mental State Examination. MMSE interpretation: score of 24–30 points indicate none or some/uncertain cognitive impairment.

**Table 2 medicina-55-00737-t002:** Participants’ median (IQR) values on the World Health Organization Disability Assessment Schedule 2.0 (WHODAS 2.0).

Assessments		WHODAS 2.0					
		Cognition	Mobility	Self-Care	Getting Along	Life Activities	Total
GMFCS:	Level I–II	1.8 (1.4–2.5)	1.6 (1.4–2.4)	1.5 (1.0–2.0)	1.4 (1.0–2.0)	1.9 (1.4–2.5)	1.7 (1.3–2.2)
	Level III–IV	1.7 (1.3–2.2)	3.8 (2.9–4.6)	3.0 (2.5–4.1)	1.4 (0.9–1.8)	2.4 (1.7–3.0)	2.5 (2.1–2.7)
	*p*-value	0.40	<0.01	<0.01	0.42	0.05	<0.01
All:		1.8 (1.3–2.3)	2.2 (1.4–2.8)	1.8 (1.0–2.9)	1.4 (1.0–1.8)	2.1 (1.5–2.6)	2.0 (1.5–2.5)

WHODAS 2.0: World Health Organization Disability Assessment Schedule 2.0, GMFCS: Gross Motor Function Classification System. WHODAS 2.0 scoring (level of difficulties in domain-specific task managing): 1—none, 2—mild, 3—moderate, 4—severe, 5—extreme or cannot do.

**Table 3 medicina-55-00737-t003:** Correlation analysis between Rotterdam Transitional Profile (RTP) and World Health Organization Disability Assessment Schedule 2.0 (WHODAS 2.0).

RTP Participation Domains	WHODAS 2.0 Domains				
	Cognition	Mobility	Self-Care	Getting Along	Life Activities
Education and employment	−0.12	−0.24	−0.22	0.05	0.14
Finance	−0.13	−0.15	−0.30	−0.01	−0.14
Housing	−0.36	−0.25	−0.42	−0.20	−0.34
Leisure (social activities)	−0.48	−0.34	−0.27	−0.19	−0.30
Intimate relationships	−0.29	−0.34	−0.44	−0.10	−0.28
Sexuality	−0.35	−0.30	−0.46	−0.19	−0.33
Transportation	−0.35	−0.64	−0.75	−0.22	−0.45

*p* < 0.001 for *r_s_* ≥ |±0.40|. RTP: Rotterdam Transition Profile, WHODAS 2.0: World Health Organization Disability Assessment Schedule 2.0.

**Table 4 medicina-55-00737-t004:** Logistic binary regression with domains of the Rotterdam Transitional Profile (RTP) and the World Health Organization Disability Assessment Schedule 2.0 (WHODAS 2.0).

RTP Participation Domains	WHODAS 2.0 Domains ^a^							
	Cognition		Mobility		Self-Care		Life Activities	
Leisure (social activities)	OR	95% CI	−		−		−	
	8.1 *	2.6–24.8						
Intimate relationships	−		−		OR	95% CI	−	
					7.5 ***	0.9–61.8		
Sexuality	−		−		OR	95% CI	−	
					9.4 **	1.2–77.0		
Transportation	−		OR	95% CI	OR	95% CI	OR	95% CI
			6.9 *	2.5–18.7	53.3 *	6.7–424.0	8.4 *	3.0–23.3

* *p* < 0.001, ** *p* < 0.01, *** *p =* 0.06. ^a^ Dichotomized into a binary variable: mean value ≤2.0 and >2.0. OR: odds ratio, CI: confidence interval, RTP: Rotterdam Transition Profile, WHODAS 2.0: World Health Organization Disability Assessment Schedule 2.0. Binary logistic regression between the housing (RTP) and self-care (WHODAS 2.0) domains could not be done because of the insufficient sample size for each group.

**Table 5 medicina-55-00737-t005:** Correlation analysis between the Rotterdam Transition Profile (RTP) domains and participants’ age and Gross Motor Function Classification System (GMFCS) level.

RTP Participation Domains	Age	GMFCS
Education and employment	0.26 ***	−0.12
Finance	0.37 *	−0.01
Housing	0.31 **	−0.16
Leisure (social activities)	0.20	−0.26 ***
Intimate relationships	0.14	−0.26 ***
Sexuality	0.28 ***	−0.31 **
Transportation	0.16	−0.56 *

* *p* < 0.001, ** *p* < 0.01, *** *p* < 0.05, for *r_s_* ≥ |±0.20|. RTP: Rotterdam Transition Profile, GMFCS: Gross Motor Function Classification System.

**Table 6 medicina-55-00737-t006:** Logistic binary regression between domains of the RTP and the participants’ age and GMFCS level.

RTP Participation Domains	Age		GMFCS ^a^	
Education and employment	OR	95% CI	−	
	2.4 ***	1.0–5.5		
Finance	OR	95% CI	−	
	2.1 **	1.3–3.2		
Housing	OR	95% CI	−	
	2.7 ***	1.2–6.5		
Leisure (social activities)	−		OR	95% CI
			0.6 ***	0.4–1.0
Intimate relationships	−		OR	95% CI
			0.4 ^¥^	0.2–1.1
Sexuality	OR	95% CI	OR	95% CI
	1.6 ***	1.1–2.5	0.3 ***	0.1–0.9
Transportation	−		OR	95% CI
			0.2 *	0.1–0.5

* *p* < 0.001, ** *p* < 0.01, *** *p <* 0.05, ^¥^
*p* = 0.07. ^a^ Dichotomized into binary variable: levels III–IV and levels I–II. OR: odds ratio, CI: confidence interval, RTP: Rotterdam Transition Profile, GMFCS: Gross Motor Function Classification System.

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
