# Peer review of "Transition-Age Young Adults with Cerebral Palsy: Level of Participation and the Influencing Factors"

_medicina, 2019, doi:10.3390/medicina55110737_

Round 1

Reviewer 1 Report

The conclusions of this study are well known.

The importance and meaning of this study should be more highlighted with supportive data analysis.

Reviewer 2 Report

This is important work and is presented in a way that is understandable to readers. I would recommend that the conclusion/discussion section be expanded to include implications for practice and particularly for future research. What do we need to understand about how to support increased independence for transition-aged youth with CP? Do we need different supports/services for adults? Do we need improved transition planning/services for students with CP before they leave high school? What, if anything, makes a difference. I believe we probably don't have the research yet to draw definite conclusions for practice, but some interesting future research questions arise from the findings of this study. This discussion might be easier to develop if the introduction were expanded to highlight what we know in general about what works to increase the independence of young adults with disabilities in general, and the importance of independence in the transition to adult life. 

Reviewer 3 Report

This is an interesting study which shows that young adults with CP show low self-management skills and autonomy. The authors suggest that the "most impacting factor" (strongest predictor?) is self-care and this should be promoted to facilitate improved transition to young adulthood. 

There are significant grammatical issues throughout the paper (e.g. incomplete/incorrect sentence structure) which made it very hard to follow. My specific comments are as follows:

Abstract

overall, too long. Try to limit to 250-300 words. It needs to be catchy and entice the reader to continue reading the whole manuscript Rationale  for the study is not clear – summarize in 1 sentence what has already been done on this topic and why is your study important Need to write in clear and full sentences e.g. “The aim of this study was to identify facilitators to participation among young adults with cerebral palsy (CP) in the context of transition to adult life” Summarize top 3-5 findings – way too many reported in abstract

Introduction

More detail needed here Need more robust evaluation of the literature in this area building to a clear rationale for why this study is important and fills a clear gap It is not clear what is meant by “therefore, much emphasis has now been put on transition process to adult life and participation” What is the transition process you are referring to? Transition from pediatric to adult care? Developmental transition to adulthood? Etc

Methods

good description of methods with sufficient level of detail

Consider making it easier for the reader by using subheadings – e.g. study design, setting, population, questionnaires, statistical analyses etc.

Results – many statistical tests were done. Clearly indicate what the primary outcome was and the secondary ones aswell. The rationale for this should be described in the introduction.

Discussion 

Although this is the strongest section of the paper, it needs to be made more concise. The differences you describe are between minors (16-17 years) and adults (18-21 years) – however, there is not much difference between a 17 year old and 18 year old. Please describe what you mean by “although it might seem obvious” as I would disagree that it is not obvious given the minor age category you used.

Reviewer 4 Report

1. The article was fully understandable. However there were some words that were incorrect English or an incorrect term

In nearly every instance ‘deficit’ should be replaced by ‘impairment’

‘Independency’ should be replaced by ‘independence’

There is one instance of ‘it’s’. This should be ‘its’  ie like ‘his’ or ‘her’, not ‘it is’

Impacting and impactful are not good English as used in this article; better to say factors that influence participation

Page 1 line 41  replace ‘previously proven’ with ‘shown’. Same change page 8 line 171

Page 2 lin86. Replace ‘level of assistant necessity’ with ‘level of assistance needed’

Page 4 line 123  ‘percentage’, not ‘percentual’

Also the whole text should be carefully re-read because there are a number of places where a word has been omitted by mistake

2. The Rotterdam transition Profile does not have an overall score. An overall score does not mean much for such a multi-dimensional concept and I recommend that all reference to it be removed?  So I think the correlation analysis Page 6 line 142 and Table 4 should be omitted – as should other analyses which use a total score.

The developers of the instrument did not construct a total score.

Dev Med Child Neurol. 2009 Jan;51(1):53-62. doi: 10.1111/j.1469-8749.2008.03115.x. Transition to adulthood: validation of the Rotterdam Transition Profile for young adults with cerebral palsy and normal intelligence. Donkervoort M1, Wiegerink DJ, van Meeteren J, Stam HJ, Roebroeck ME; Transition Research Group South West Netherlands.

3. Page 2 Line67. Why were the three health care domains not used?

4. How were patients identified? Page 2 line 52

Very little information is given about how the participants were identified. The authors should comment on the extent to which the participants might be deemed representative of the population studied. Therefore:

The sample from which they were approached – was that sample representative?

Then what happened to those approached;  non-contactable, refusals, dropped out.

Round 2

Reviewer 4 Report

The authors have still failed to replace 'indpendency' with 'independence' in two places.

The authors have now also introduced the word 'autonomy' This does not mean the same as independence and the authors should consider carefully if they really mean 'autonomy' or whether in fact it should be 'independence'.

('Autonomy means you are in control of events, even though you may be dependent on other people or things to achieve them).

The authors have failed in Figure 1 to remove the bottom 2 lines  relating to 'Total'.

Page 2 line 58. Replace 'where searched' with 'were sought'
